# Comprehensive Analysis of Metabolites in Brews Prepared from Naturally and Technologically Treated Coffee Beans

**DOI:** 10.3390/antiox12010095

**Published:** 2022-12-30

**Authors:** Magdalena Jeszka-Skowron, Robert Frankowski, Agnieszka Zgoła-Grześkowiak, Julia Płatkiewicz

**Affiliations:** Institute of Chemistry and Technical Electrochemistry, Poznan University of Technology, Berdychowo 4, 60-965 Poznań, Poland

**Keywords:** coffee, alkaloids, caffeoylquinic acids, nicotinic acid, phenolic acids, indolamines

## Abstract

Coffee is one of the most popular beverages in the world. Therefore, this study analyzed 49 coffee samples of Arabica and Robusta species of different geographical origins and the treatment of beans including three degrees of roasting with the use of LC-MS/MS. This is the first study to present a comprehensive analysis of Kopi Luwak coffee brew metabolites in comparison to fully washed coffees and the drying post-harvest treatment of Arabica or Robusta coffee brews. Kopi Luwak showed higher levels of caffeine and theophylline in comparison to the analyzed washed and unwashed Arabica coffees, as well as a different proportion of caffeoylquinic isomers. There was no difference between Kopi Luwak and other Arabica coffees in terms of the concentration of vitamin B_3_, amines, and phenolic acids. This was confirmed in PCA. The steaming and roasting of beans as well as the addition of black beans influence the concentration of 4-CQA and the nicotinic, ferulic, and quinic acids content.

## 1. Introduction

Coffee is one of the most popular beverages prepared and drunk in homes and restaurants. Apart from its unique aroma and flavor and stimulative properties, drinking coffee (from two to four cups per day) has other potential positive effects associated with reducing the incidence of stroke cases [1] and lowering the risk of mortality [2].

Green coffee beans contain proteins and polysaccharides, caffeine and trigonelline, chlorogenic acids (CGAs), and other bioactive compounds and elements. The roasting process influences the physical changes and chemical reactions such as the Maillard and Strecker reactions and the production of brown pigments from monosaccharides [3]. During the roasting process at 180 °C, the “first crack“ shows that the light roasting of beans is finished. It was reported that transformations such as isomerization (acyl migration) or hydrolysis of the ester bond take place in the CGAs during the early stages of roasting [4]. Then, chemical transformations such as the decarboxylation of cinnamoyl moieties to produce a number of phenylindanes, epimerization at the quinic acid, and dehydration to produce cyclohexenes and lactones occur [5]. The “second crack” appears when the dark coffee roasting process is needed. Proteins, polysaccharides, chlorogenic acids, and the trigonelline content decrease, other substances such as volatile compounds are formed, and nicotinic acid and nicotinamide levels are increased [6,7].

There are different processes, such as steaming, decaffeination, and natural fermentation, that are used to treat coffee beans. Kopi Luwak coffee is a specialty coffee, in which beans are fermented in the civet’s gut. The Asian palm civet (*Paradoxurus hermaphroditus*) is a small mammal native to Southern and Eastern Asia, especially Java, Sumatra, and Sulawesi. The animal eats the best-flavored cherries and excretes them [8]. After the Luwak’s feces with beans are collected, the beans are brought out and cleaned, usually wet-fermented, and then sun-dried.

From the bioactive group of compounds, coffee beans contain mainly chlorogenic acids, which account for almost 90% of all phenolic compounds, including three main isomers: 5-CQA, 4-CQA and 3-CQA and/or feruloylquinic (FQA) and dicaffeoylquinic compounds. The level of total chlorogenic acids can vary largely, and the range is usually from 50–200 mg 100 mL^−1^ [9] to even 812 mg 100 mL^−1^ (considering unusually concentrated espresso coffees) [9,10]. “[The] [p]ercent distribution of CGA compounds in coffee brew, in order of abundance, is, on average: 5-CQA (41–48%), 4-CQA (20–25%), 3-CQA (17–20%), 5-FQA (4–8%), 4-FQA (2–5%), 3-FQA (1–4%), 3,4-diCQA (1.0–2.5%), 3,5-diCQA (1.0–1.5%), 4,5-diCQA (~1%), others (<1%)” [11]. Other phenolic compounds can also be determined in lower amounts in green and/or roasted coffee brews such as flavonoids ((-)-gallocatechin 3-gallate, (-)-catechins 3-gallate, (+)-catechin and rutin [12,13]), isoflavones, and lignans [14].

Methylxanthines are the main alkaloids in coffee. Caffeine is a dominating compound in this group. Trigonelline (N-methyl nicotinic acid), the other important alkaloid, can be determined in coffee on a similar level as caffeine in some cases [15].

Bioactive amines from the indoleamine group such as serotonin and melatonin are mainly determined in green coffee beans [16]. These compounds act as neurotransmitters and are involved in cellular processes in the human organism that are induced in the brain. Serotonin (5-hydroxytryptamine) controls blood pressure, smooth muscle stimulation, and sensory threshold rises and inhibits gastric secretion [17]. L-theanine is a non-protein amino acid that is biosynthesized from glutamic acid and ethylamine by the enzyme theanine synthetase [18].

In our previous studies, we have established that coffee beans processing (including roasting, steaming and decaffeination) influenced the content of chlorogenic acids, caffeine, antioxidant activity, and metals such as Cu and Mn in coffee beans and brews [12,19,20]. These processes also had an impact on the nicotinic acid and nicotinamide contents [7].

The compounds with a high antioxidant activity in coffee are mainly chlorogenic acids, and Robusta coffees had higher antioxidant activities in the Folin–Ciocalteu (F–C) assay due to a higher level of the main chlorogenic acids content (176 mg L^−1^—average for Robusta coffees) compared to that of Arabica coffees (153 mg L^−1^—average for Arabica coffees) [20]. In addition, a high (0.807) correlation between F–C and CUPRAC assays of green coffee brews with the use of Pearson’s linear correlation coefficient was obtained [19]. A positive correlation between the sum of three major chlorogenic acids and F–C (0.667) in green coffee brews was also found [19].

Obtaining a high-quality coffee brew from coffee beans is not only related to the species of coffee, the roasting of the coffee beans, and their origin but also to the processing of the coffee fruit immediately after harvesting. A distinction is made between the dry method (drying whole fruits), the wet method, and the hybrid methods. When the process parameters are properly controlled, the wet method generates fewer low-quality coffee beans, allowing for the production of coffee with better aromatic properties compared to those produced by the dry method [21].

The aim of this study was to compare the concentrations of six isomers of chlorogenic acid, alkaloids, nicotinic acid and nicotinamide, serotonin and melatonin, as well as ferulic, caffeic, *p*-coumaric, and quinic acids in brews from unfermented and fermented coffee beans with the use of LC-MS/MS. Arabica coffees such as Brazilian Mogiana (dry coffee beans), Uganda Bugishu, Guatemala, Papua New Guinea (fully washed beans), Kopi Luwak, as well as Robusta coffee beans (dry beans) from Brazil or Vietnam were analyzed. In addition, Principal Component Analysis (PCA) was used to show the influence of different technologies (steaming, decaffeination, and roasting degree) and different qualities of beans (black and broken beans) on the bioactive compounds determined in coffee brews.

## 2. Materials and Methods

### 2.1. Chemicals

All analytical standards were purchased from Merck/Sigma-Aldrich (Steinheim, Germany). MS-grade acetonitrile, methanol, and formic acid were also obtained from Sigma-Aldrich (Steinheim, Germany). High-purity water—deionized (DEMIWA 5 ROSA, Watek, Ledeč nad Sázavou, Czech Republic ) and doubly distilled (quartz apparatus, Bi18, Heraeus, Hanau, Germany)—was used throughout the research. The resistivity of the water was 18 MΩ cm.

### 2.2. Sample Preparation

Sixteen samples of green coffee beans of different origins of *Coffea arabica* (two samples from Brazil (Brazil Arabica MTGB, Brazil Arabica Mogiana TF), Colombia Arabica Excelso, Guatemala Arabica SHB, Uganda Arabica Bugishu, Uganda Arabica Drugar) and Coffea robusta (Brazil Robusta Conillon 6/7 200 def., Brazil Robusta Conillon 5/6 100 def., Uganda Robusta screen 12, Uganda Robusta Black Beans, Tanzania Robusta sc14, Vietnam Robusta gr. 2, 5% BB, Vietnam Robusta screen 16 clean), as well as Vietnam Robusta processed beans (gr. 2, 5% BB Steamed (3 bar pressure for 30 minutes)) and Vietnam Robusta (decaffeinated by dichloromethane), were obtained from a producer (Strauss Café, Tarnowo Podgórne, Poland) (Table 1). The geographical origins of the samples and their types were established by the provider. The moisture content of coffee beans was below 12%. All coffees were roasted at three degrees (light—R1, medium—R2, and dark-roasted—R3): first crack, city to full-city, and second crack. Kopi Luwak coffee (light-medium-roasted degree) from Bali, Indonesia was purchased at an Indonesian market.

For the homogenization of coffee, it was cooled down with the use of liquid nitrogen and ground for 1 min in a high-speed blender equipped with a stainless-steel jar (model A11 basic; IKA Works GmbH & Co.; Staufen; Germany) to prevent the loss of bioactive components. All samples were stored at −20 °C prior to chemical analysis. Before the LC-MS/MS analysis, the solution was filtered through a 0.2 µm polytetrafluoroethylene syringe filter from Agilent Technologies (Santa Clara; CA; USA) and diluted with 7% acetonitrile, depending on the compounds to be determined, in following the ratios: 1:5 for nicotinic acid, nicotinamide, and bioactive amines; 1:100 or 1000 for caffeine and other organic acids.

### 2.3. LC-MS/MS

The determination of 19 bioactive compounds—6 isomers of chlorogenic acid (5-O-caffeoylquinic acid; 4-O-caffeoylquinic acid; 3-O-caffeoylquinic acid; 4,5-dicaffeoylquinic acid; 3,5-dicaffeoylquinic acid; 3,4-dicaffeoylquinic acid), serotonin, melatonin, theanine, nicotinamide, nicotinic acid, trigonelline, theophylline, theobromine, and caffeine, as well as ferulic acid, caffeic acid, *p*-coumaric acid, and quinic acid—was accomplished using the UltiMate 3000 RSLC chromatographic system from Dionex (Sunnyvale, CA, USA). A total of 5 µL of the samples was injected into an Ascentis Express 90A RP-Amide column (100 mm x 2.1 mm I.D.; 2.7 µm) from Supelco, Merck (Darmstadt, Germany), maintained at 35 °C. The analysis was performed in a gradient of two mobile phases: 0.1% formic acid in water and acetonitrile flowing at 0.3 mL min^−1^ (Appendix A) in a gradient. The effluent from the analytical column was introduced into the electrospray ionization source of the API 4000 QTRAP triple quadrupole mass spectrometer from AB Sciex (Foster City, CA, USA) operating in positive ion mode, as described in Appendix A. The dwell time for each transition in the multiple reaction monitoring mode was set to 50 ms. The compound-dependent detection settings are given in Appendix A. The LC-MS/MS parameters for the analyzed compounds, such as the retention time, the limit of detection (LOD), and the limit of quantification (LOQ), can be found in Appendix A.

### 2.4. Optimization of Green and Roasted Arabica and Robusta Coffee Brewing Conditions

The optimization of coffee brewing for Arabica (Mogiana, Brazil) and Robusta (Vietnam) green and roasted coffee beans (medium degree) was accomplished. A total of 0.5 g of milled beans was extracted by 50 mL of distilled water at the proper temperature (60, 70, 80, 90, and 95–100 °C) and time (2, 4, 6, 8, and 10 min.) (Appendix A; Appendix A). The optimal parameters were: boiling water (95–100 °C) and extraction for 10 minutes. After cooling the solution to room temperature, it was decanted.

### 2.5. Statistical Analysis

The results were expressed as the mean ± standard deviation (at least three replicates). The central composite design (CCD) was used for the optimization of experiments (the factorial design contained four factorial points, four axial points, and five central points). The adequacy of the models was determined by evaluating the lack of fit, the coefficient of determination R2, and the adjusted R2, and the Fisher test value (F value) was achieved by the analysis of variance (ANOVA). The Principal Component Analysis (PCA) between unprocessed and processed samples was employed. The PCA of the results was conducted to visualize and elucidate the association between the samples and constituents. The experimental data were analyzed using the Statistica 13.0 program (StatSoft Inc., Tulsa, OK, USA).

## 3. Results and Discussion

### 3.1. Determination of Chlorogenic Acids and Their Isomers

The six isomers of the caffeoylquinic acids in all coffee brews were determined, and the dominant acid was 5-O-caffeoylquinic acid, followed by 4-O-caffeoylquinic acid and 3-O-caffeoylquinic acid (Figure 1 and Figure 2). 4,5-dicaffeoylquinic acid, 3,5-dicaffeoylquinic acid, and 3,4-dicaffeoylquinic acid were also found in all analyzed coffees. 5-CQA was the main compound in all green unprocessed beans (from 72% to 83% in the Arabica coffees and from 41 to 80% in the Robusta coffees of the total analyzed CGAs), and the amount of this compound decreased after light roasting. In medium-roasted coffees, the levels of other isomers increased, especially 4-CQA, and in dark-roasted coffees, they usually decreased (Figure 1 and Figure 2). The lowest levels of 5-CQA and other isomers as well as the total analyzed CGAs were found in most of the dark-roasted coffees.

Similar results were previously reported, especially the level of CGAs for very dark coffees (French or Italian roast), reaching above 90% of 5-CQA among all isomers, while 3,4-diCQA was not detected [22]. This is a consequence of the thermal breakage of carbon–carbon covalent bonds, resulting in isomerization in the initial roasting stages and epimerization, lactonization, and degradation in the later stages [22].

Among the Arabica green coffees, Uganda Bugishu (AUB) showed the highest level of total determined chlorogenic acids: 344 mg g^−1^ (Figure 1). After roasting, the proportion of chlorogenic acids changes. While 5-CQA is decreased, the levels of 4-CQA are usually increased after light and medium roasting, which is especially visible in AUB and AG coffee brews. Dark-roasted coffees (R3), both Arabica and Robusta species, usually showed the lowest levels of 5-CQA, 4-CQA, and 3-CQA (Figure 1 and Figure 2). Similar results were obtained for coffee brews prepared from Arabica beans from the Dominican Republic, Papua New Guinea, and Ethiopia [23]. The decrease in the three main chlorogenic acids was similar to that in previous research, but the proportion among 4,5-dicaffeoylquinic acid, 3,5-dicaffeoylquinic acid, and 3,4-dicaffeoylquinic acid was different [24,25].

Generally, green beans possessed lower levels of 3-CQA and 4-CQA than the brews prepared from light-roasted beans. This is connected with acyl migration via the formation of an ortho-ester intermediate [26]. Furthermore, the amount of 3,5-diCQA was higher than 3,4-diCQA in most green unprocessed coffees, which changed after roasting.

Liang et al. [23] found that green Arabica coffee samples had high contents of 5-CQA and dicaffeoylquinic acids and more intracellular antioxidant activity. Relatively high contents of 3-CQA and 4-CQA were found for light-roasted coffee samples. This was accompanied by a powerful peroxyl radical observed in the ORAC assay. On the other hand, medium- and dark-roasted coffee samples had a content of high-molecular-weight melanoidin clustered with the TEMPO radical scavenging capacity. Thus, both green and roasted Arabica coffee beans sourced from five geographic locations showed antioxidant activities.

Recently, the determination of the individual CGAs isomers, color parameters, and antioxidant activity of Thai Arabica green coffees prepared by different drying treatments was shown [27]. The drying process affected the contents of CGAs, TPC, and FRAP, the % inhibition by DPPH, and the color parameters of green coffees, and heat pump drying at 50 °C can be an alternative and possibly an advantage over sun-drying for preserving the bioactive compounds of green coffee.

The post-harvest treatment of Uganda Arabica coffees resulted in the differences between the analyzed CQAs compounds (Figure 1). A higher amount of 4-CQA and 3,5-CQA in green and roasted (on all degrees) Bugishu coffee compared to Drugar coffee was found. The highest level of the total CGA compounds in Uganda Bugishu fully washed coffee was also determined. The effect of the post-harvest drying practice on the CGA isomer profiles of Arabica green coffee with high positive coefficients (>0.7) for 5-CQA, 5-FQA (5-feruloylquinic acid), 4-FQA (4-feruloylquinic acid), 3,4-diCQA, 4,5-diCQA, and total CGAs was found [27].

Kopi Luwak Arabica coffee showed the lowest level of chlorogenic acids isomers (90.7 mg g g^−1^; sum of all analyzed compounds) in contrast to all Arabica green and roasted-at-three-degrees coffees tested in the present study, especially washed and light-roasted Arabica coffee from Uganda, Bugishu (340.5 mg g^−1^) (Figure 1). Apart from roasting, the level of 5-CQA in Kopi Luwak could be lower than regular in the coffee beans due to the metabolism by bacteria in the civet digestive system. It has been reported previously that 5-CQA was metabolized by bacteria in the large intestine cecum of Wistar rats [28]. Similarly, the amount of total CGAs in Kopi Luwak coffee from Indonesia and Jacu bird depulped from Brazil was 1.5-fold lower (on average) than that from Columbia or Peru regular coffees [24].

It is important to notice that two unwashed coffees from Brazil showed differences between the CGAs contents; Mogiana type New York (no black beans allowed) possessed a higher level of the compounds in green and roasted coffee bean brews than the Brazilian medium-to-good beans, as expected (Figure 2).

It was recently found that not only the roasting process but also the storage can influence the chlorogenic acids determined with the use of HPLC (as chlorogenic acids without information about isomers), as found in Brazilian Arabica coffees [13]. After 12 months of storage, the level of chlorogenic acids decreased by more than 90%. In addition, the levels of chlorogenic acids in conventional and organic coffee were similar.

The total analyzed chlorogenic acid content in Robusta coffee brews was usually higher than that in Arabica coffee brews (Figure 1 and Figure 2), which was reported previously [11]. The highest level of these compounds was determined in the Brazil Conillon green coffee brew 5/6—800 mg g^−1^, followed by the Vietnamese decaf coffee brew—551 mg g^−1^ (Figure 2). Recently, it was found that decaffeination with supercritical carbon dioxide (sCO_2_) showed the highest level of total CGA and CGL—7.2% for Colombian Arabica coffee and 7.8% for Indonesian Robusta—in comparison to the water and dichloromethane (DCM) method [24].

Robusta coffees from Uganda and Vietnam with black beans showed a lower level of 5-CQA compared to both types of Brazilian black bean coffees (5/6 and 6/7). Interestingly, these Brazilian Conilon coffees of lower-quality beans showed a high level of CGAs (Figure 2).

Steamed black beans coffee from Vietnam showed a higher level of total chlorogenic content in comparison to unsteamed black beans coffee from Vietnam (Figure 2). Furthermore, the level of 5-CQA was 1.8-fold lower, and the amount of 4-CQA increased 6.7-fold, respectively. This phenomenon is due to the special treatment of coffee, which can increase the quality of coffee.

The highest level of the total determined CGAs was obtained for medium-roasted coffee from Vietnam with black beans (RVBBS = 459.6 mg g^−1^) (Figure 2). In contrast, RVBB light- and medium-roasted coffees possessed total CGAs contents of 329.7 mg g^−1^ and 333.6 mg g^−1^. In addition, the level of CGAs in RV was higher for green beans and light-roasted beans in comparison to RVBB, but in medium- and dark-roasted RV, it was two- and three-fold lower. Coffee brew (Robusta) from Uganda (RU) showed a higher level of total CGAs compared to Uganda (RUBB) black beans coffee in brews prepared from green and roasted-on-all-degrees coffee beans. This could be due to the lower quality of black beans coffee. To the best of our knowledge, this is the first report on CGA content in black beans coffee from Vietnam and the comparison with steamed coffee beans.

### 3.2. Determination of Methylxanthines and Trigonelline

Caffeine was the major alkaloid in all analyzed samples. For Arabica coffees, the highest level of the compound was found in the Kopi Luwak coffee brew (31.6 mg g^−1^), followed by the Brazilian Mogiana coffee brew (29.4 mg g^−1^) (Figure 3). In the case of Robusta coffees, the highest amount was found for Robusta Conilon 5/6 coffee (51.6 mg g^−1^), and the level was rather stable during the roasting of coffee in most of the samples (Figure 4).

On the other hand, the highest amount of trigonelline was determined in Arabica Brazil coffees (11.5 mg g^−1^ for Mogiana coffee and 11.2 mg g^−1^ for MTGB coffee), and during roasting, the amount of this compound was decreased by even more than three-fold. The steaming of coffee beans reduced the level of trigonelline, and the amount of this compound is lower for steamed and roasted coffee (average: 3.5 mg g^−1^ for all degrees of roasting) in comparison to that of unsteamed and roasted coffee.

The level of theophylline in Arabica coffees was the highest for Kopi Luwak (0.044 mg g^−1^) in comparison to the washed coffees (from 0.001 mg g^−1^ for Uganda Bugishu to 0.012 mg g^−1^ for Uganda Drugar) and higher than the concentration of theobromine (Figure 3 and Figure 4). On the other hand, the concentrations of theophylline in other Arabica coffees were considerably lower than those of theobromine. In Robusta coffees, the green coffee brews RVBB and RUBB showed the highest content of the compound—even higher than that of theobromine (Figure 4). The steaming of coffee increased the level of theobromine in the RVBB green coffee brew from 0.0089 mg g^−1^ to 0.0235 mg g^−1^ (2.6-fold higher), and in steamed and roasted coffee, it was 0.038 mg g^−1^. The decaffeination process decreased the level of theophylline, on average, by 10-fold but did not change the amount of theobromine.

The level of caffeine can be in the range from 0.2 to 1.1 mg g^−1^ for decaffeinated coffees and from 8.4 to 50.9 mg g^−1^ for regular roasted coffees [7,15,29]. Some differences in the caffeine content between raw and roasted coffee beans could be related to the loss of water, carbon dioxide, and volatile compounds during the roasting process, which share the composition of raw coffee.

In the case of theophylline, it is usually in the lowest amounts in the group of methylxanthines—in the range from 3.7 to 15.7 mg g^−1^ in roasted and decaffeinated coffee brews [29].

The trigonelline content in green coffee brews ranges significantly from 2.7 to 36.9 mg g^−1^ [7,15,29]. The difference between the dry and wet methods is due to the wet method involving the soaking process in water for about 12 h to remove the mucus. This soaking process possibly dissolved trigonelline [30].

### 3.3. Determination of Nicotinic Acid and Nicotinamide

After roasting, the level of trigonelline is decreased, and, therefore, the amounts of nicotinic acid (NA) and nicotinamide (NAM) are increased. Dark-roasted coffees contain the highest level of nicotinic acid: 0.217 mg g^−1^ for Arabica Brazil Mogiana (dark-roasted, second crack) and 0.203 mg g^−1^ for Robusta Vietnam decaffeinated coffees (Figure 5 and Figure 6). Therefore, an 87-fold increase in the compound for the Brazil Mogiana coffee brew was observed. Green coffee brews contained only 0.0016 mg g^−1^ to 0.0033 mg g^−1^ of nicotinic acid for Arabica coffees, and this was 0.0027 to 0.0061 mg g^−1^ for Robusta unprocessed beans (Figure 5 and Figure 6).

The decaffeination process increased the level of nicotinic acid by about 2.0-fold in green coffee beans (RVD) in comparison to Vietnam caffeinated coffee beans (RV), and after roasting (all degrees), the amount was decreased (Figure 6). However, after roasting, the level of the steaming of the coffee beans of Robusta Vietnam Black Beans coffee as well as the steaming and roasting on all three degrees increased the level of nicotinic acid and nicotinamide in comparison to Robusta Vietnam Black Beans. Both the green and roasted coffees were on the third degree of roasting. On the other hand, the lowest level of nicotinamide was found for green coffee beans (the same as that of nicotinic acid), but after roasting, it was from three- to eight-fold higher without the influence of the degree of roasting. The steaming of coffee beans increased (four-fold) the level of nicotinamide in coffee brews.

Nicotinic acid (with trigonelline) is usually determined between other compounds such as sucrose or chlorogenic acids with the use of HPLC or LC-MS [15,31]. The analysis of nicotinamide in coffees is rather rare [7,29]. Perrone et al. [15] found that the amount of nicotinic acid in green coffee beans was below the limit of detection and achieved an LOD = 15.5 ng mL^−1^ in comparison to that of our results (0.06 ng mL^−1^) (Appendix A). We recently found that Arabica and Robusta green coffee brews contain nicotinic acid at similar levels, from 3.20 to 4.81 μg g^−1^ and from 4.01 to 6.72 μg g^−1^, respectively, depending on the coffee origin. This is also true of nicotinamide—from 0.31 to 0.88 μg g^−1^ and from 0.30 to 1.66 μg g^−1^, respectively [7]—and similar levels were also found in other studies [15,29,32]. After the light roasting process, the level of NA increased in Arabica coffees from 4.7-fold to 23.1-fold, and in Robusta coffees, it increased from 3.3-fold to 1.8-fold. NAM changed in Arabica coffees from 1.5-fold to 4.1-fold, and in Robusta coffees, it changed from 1.1-fold to 3.2-fold [7].

### 3.4. Determination of Bioactive Amines: Theanine, Serotonin, and Melatonin

Coffee beans and brews can be a source of amines. It was found that the highest level of serotonin determined in all green coffees was from 0.010 mg g^−1^ to 0.031 mg g^−1^ for Arabica coffees and from 0.016 to 0.0281 mg g^−1^ for Robusta coffees (Table 2). The decaffeination, steaming, and roasting of coffee beans at all degrees influence the level of serotonin, and its amount decreased by 21-fold to even 100-fold. A similar effect of the processes was found for theanine, whose concentration decreased from a few µg g^−1^ to a fraction of µg g^−1^ (Table 2).

The melatonin content in all green, decaffeinated, steamed, and roasted Arabica (fermented or unwashed coffees) and Robusta coffees was determined to be of very low amounts—from 1.5 to 12 ng g^−1^. Previously, the melatonin content in green coffee beans was at an even lower level of 39.0 ± 6.5 pg g^−1^, and that of melatonin isomer was 1.2 ± 0.1 pg g^−1^ [33]. Amines can be thermolabile, and immature coffee processed by depulped, natural, rest-in-water, or rest-dry methods had no influence on the level of serotonin [34].

Recently, it was found that the serotonin content in different coffee samples was from 0.001 to 0.006 mg g^−1^, which is 10-fold higher than that in our research [35]. The level of serotonin in coffee beans after the amine extraction, cleanup, and dansylation of the raw coffee beans was from 1.82 mg g^−1^ for Columbian Arabica to 3.21 mg g^−1^ for Angolan Robusta coffee beans [36]. Regarding free serotonin, 0.23 ± 1.2 was found for Arabica, and 0.33 ± 0.8 mg g^−1^ was found for Robusta coffee raw beans [37]. The differences in the results originated from the different extractions of the samples. Principal component analysis (PCA) had not shown the differences between the wet and dry coffee treatments and the serotonin content [37]. In addition, ground coffee samples (purchased from the market in Izmir, Turkey) possessed a higher level of serotonin: 0.107 mg g^−1^ [38]. In the brewed Turkish prepared coffees from the Turkish market, the mean value of serotonin was 6.7 mg L^−1^ [38].

Apart from the serotonin aglycone, the serotonins 5-O-β-glucoside and tryptophan-N1-glucoside were determined in the green coffee beans [39]. The serotonin 5-O-β-glucoside was found in all varieties of Robusta (0.0031 ± 0.0004 mg g^−1^) and Arabica (0.0054 ± 0.0008 mg g^−1^) coffee species.

### 3.5. Determination of Quinic Acid and Other Phenolic Acids (Ferulic, Caffeic, and P-coumaric Acids)

The amount of quinic acid in Arabica coffee was the lowest in green coffee beans: 1.93 mg g^−1^ in Papua New Guinea coffee (Figure 7). After roasting, its concentration increased by more than two-fold—for example, for Arabica Drugar coffee. The level of quinic acid increased after roasting due to the decreasing content of chlorogenic acid isomers (esters of quinic and caffeic acids). Interestingly, the level of quinic acid was stable after steaming coffee and then roasting at the third degree in Robusta coffee (Figure 8). The highest content of the compound was found for Brazilian 6/7 Robusta coffee (17.3 mg g^−1^) at the third degree of roasting.

The highest amounts of ferulic acid and caffeic acid were found in the Tanzanian Robusta coffee brew: 0.548 mg g^−1^ and 0.242 mg g^−1^, respectively (Figure 8). A similar level of caffeic acid was determined in the green beans of Uganda Robusta Black Beans coffee: 0.249 mg g^−1^. The roasting process decreased this compound, especially at the third degree, inversely to quinic acid.

The decaffeination process increased the level of ferulic acid (3.1-fold) and quinic acid (1.5-fold) for green coffee beans (Figure 8). This process did not influence the *p*-coumaric acid content. For steamed coffee, the level of *p*-coumaric acid was lower by 4.7-fold (RVBBS) in comparison to RVBB green coffee beans.

It was recently found that, with the degree of roasting, the level of caffeic acid and the amount of it in Brazilian Arabica coffee beans decrease [13]. The results presented in this present study confirm these findings.

The level of phenolic acids such as *p*-coumaric and caffeic acids decreased during the roasting process of coffee beans. Similar findings were described by Górnaś et al. [40]. In addition, they have found that medium-roasted coffee with the highest total phenolic acids possesses the highest antioxidant activity, as measured by spectrophotometric assays. Significant correlations between browned compounds, trigonelline, 5-caffeoylquinic acid and caffeic acid contents, and the antioxidant activity measured by the DPPH and redox potential methods (total phenolic compound test, TRAP, FRAP) were presented using statistical analyses [40].

### 3.6. Principal Component Analysis

A principal component analysis (PCA) was performed to explain the differences between the samples of coffees in terms of the genus, roasting degree, decaffeination, steaming process, and coffee beans with the addition of black beans. The first two components’ biplot (Figure 9) showed that PC1 (axis 1) negatively correlates with a few variables. Observations with the largest negative coordinate on the horizontal axis correspond to the most important compounds such as 5-CQA, with a high antioxidant activity. Along the vertical axis, quinic and nicotinic acids as well as theophylline and theobromine are in opposition with trigonelline and *p*-coumaric acid.

The PCA analysis illustrates the similarities between the green and processed samples of coffee (Figure 10). It showed some degree of separation between the green and roasted coffee samples, explaining 54.67% of the variance with two principal components. The PCA indicated that green and roasted coffee could in fact be significantly different in relation to alkaloids, chlorogenic acid isomers, and phenolic acids. Moreover, for the samples, no clear, distinct clusters were obtained (among light-, medium-, and dark-roasted beans).

## 4. Conclusions

This is the first study to show a comprehensive analysis of Kopi Luwak coffee brew metabolites in comparison to fully washed coffees and the drying post-harvest treatment of Arabica or Robusta coffee brews. Kopi Luwak showed a higher level of caffeine and theophylline in comparison to the analyzed Arabica regular coffees, as well as a different proportion of caffeoylquinic isomers. There was no difference between Luwak and the other Arabica coffees in terms of the concentration of vitamin B_3_, analyzed amines, and phenolic acids. This was confirmed in PCA. Robusta Brazilian Conilon coffee type 5/6 showed the highest levels of caffeine, nicotinamide, and quinic acid. The steaming and roasting of beans as well as black beans coffee influence the concentration of 4-CQA and the nicotinic, ferulic, and quinic acids contents.

To the best of our knowledge, this is the first study to report the CGA content in Robusta coffee from Vietnam with black beans and a comparison with steamed coffee beans. Steamed black beans coffee from Vietnam showed a higher level of total chlorogenic content in comparison to unsteamed black beans coffee from Vietnam on all degrees of roasting. The highest level of 4-CQA from the analyzed CGA was found in steamed and roasted Vietnam coffee black beans.

## Figures and Tables

**Figure 1 antioxidants-12-00095-f001:**
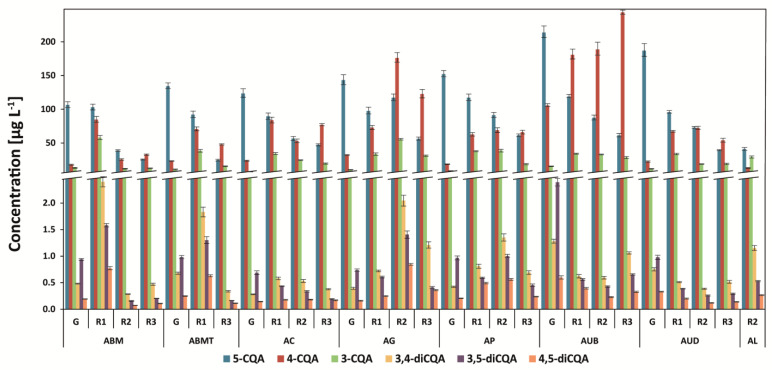
Chlorogenic acid (caffeoylquinic acids isomers—CQA) content in Arabica coffee brews. Legend: G—green coffee beans; R1—light-roasted coffee beans; R2—medium-roasted coffee beans; R3—dark-roasted; AL—Arabica Kopi Luwak; 5-CQA—5-O-caffeoylquinic acid; 4-CQA—4-O-caffeoylquinic acid; 3-CQA—3-O-caffeoylquinic acid; 3,4-diCQA—3,4-dicaffeoylquinic acid; 3,5-diCQA—3,5-dicaffeoylquinic acid; 4,5-diCQA—4,5-dicaffeoylquinic acid. Detailed information about the coffees can be found in Table 1.

**Figure 2 antioxidants-12-00095-f002:**
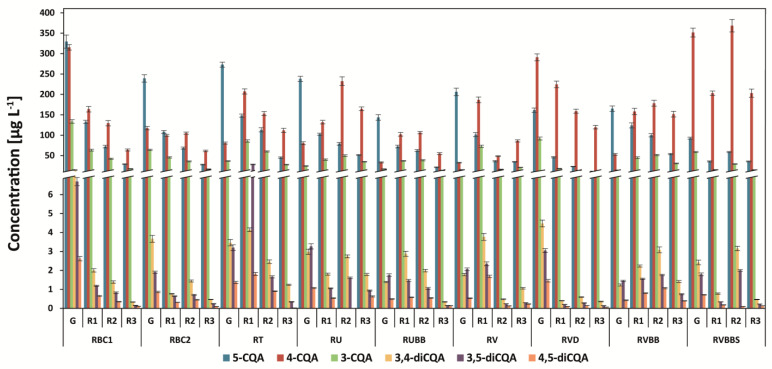
Chlorogenic acid (caffeoylquinic acids isomers—CQA) content in Robusta coffee brews. Legend: information about the coffees is provided in Figure 1.

**Figure 3 antioxidants-12-00095-f003:**
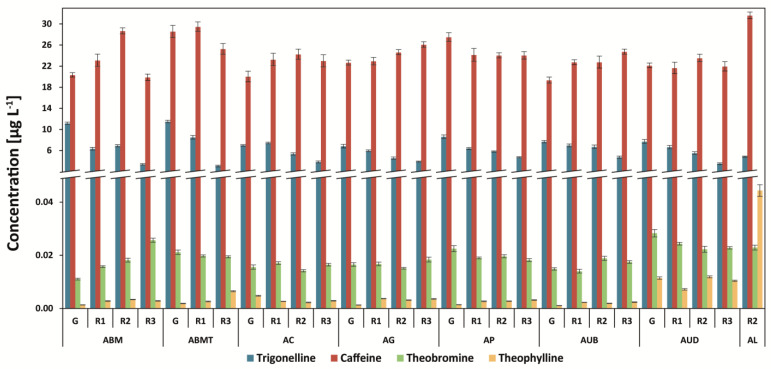
Methylxanthines and trigonelline content in Arabica coffee brews. Legend—Figure 1 capture.

**Figure 4 antioxidants-12-00095-f004:**
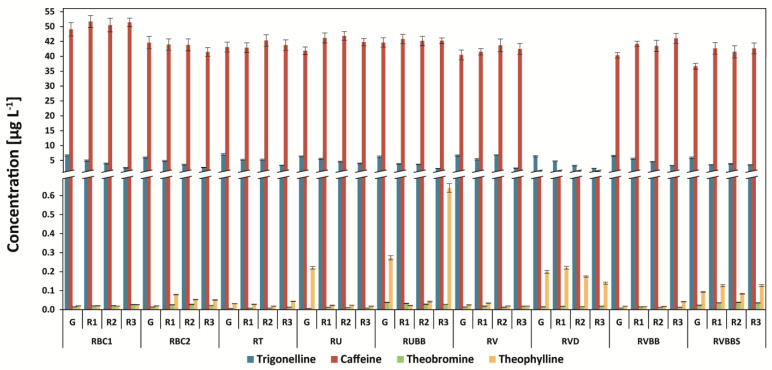
Methylxanthines and trigonelline content in Robusta coffee brews. Legend—Figure 1 capture.

**Figure 5 antioxidants-12-00095-f005:**
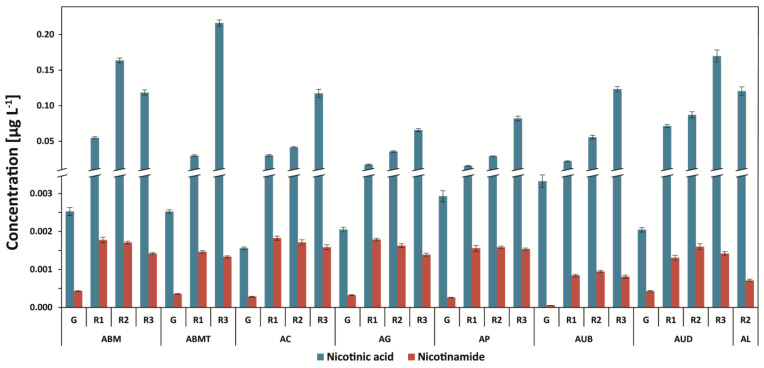
Nicotinic acid and nicotinamide content in Arabica coffee brews. Legend—Figure 1 capture.

**Figure 6 antioxidants-12-00095-f006:**
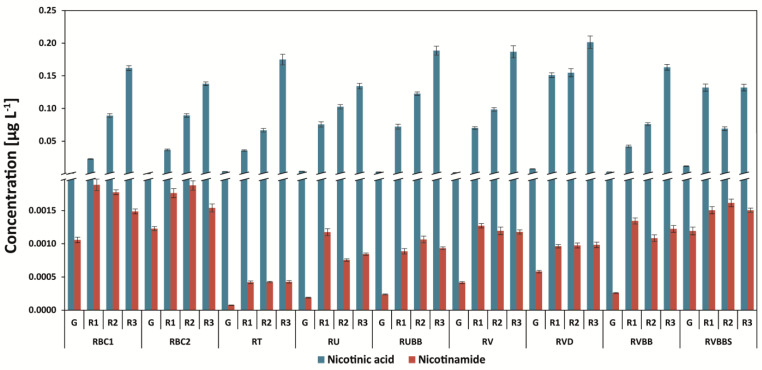
Nicotinic acid and nicotinamide content in Robusta coffee brews. Legend—Figure 1 capture.

**Figure 7 antioxidants-12-00095-f007:**
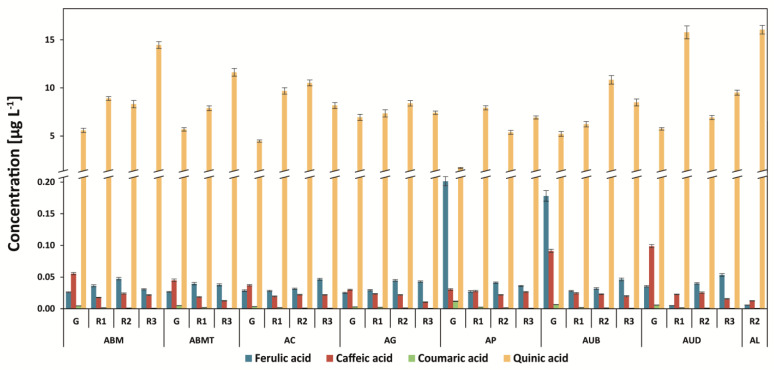
Ferulic, caffeic, *p*-coumaric, and quinic acids in Arabica coffee brews. Legend—Figure 1 capture.

**Figure 8 antioxidants-12-00095-f008:**
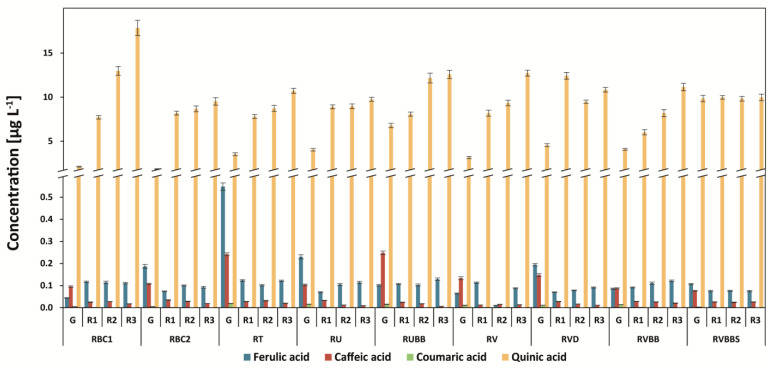
Ferulic, caffeic, *p*-coumaric, and quinic acids in Robusta coffee brews. Legend—Figure 1 capture.

**Figure 9 antioxidants-12-00095-f009:**
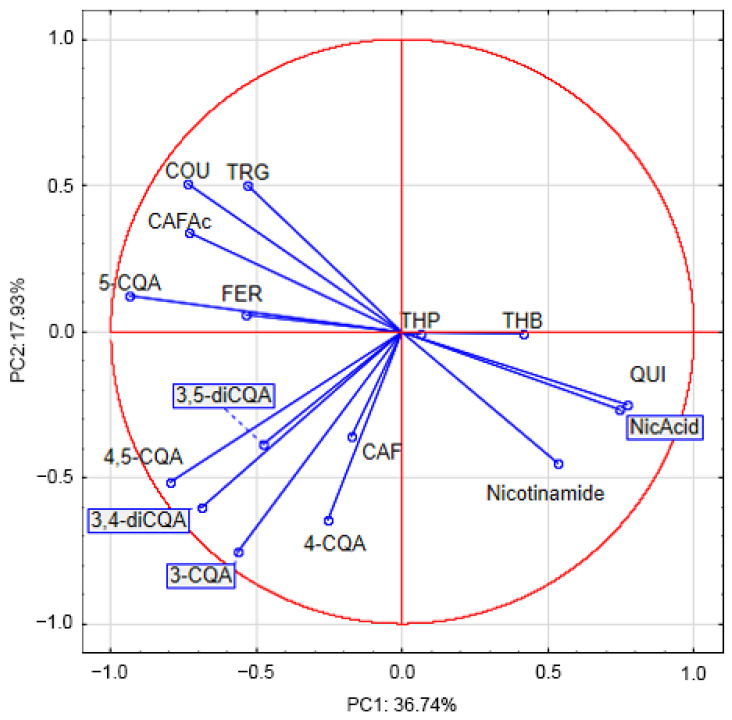
Principal Component Analysis—Projection of variables (TRG—trigonelline; COU—coumaric acid; CAFAc—caffeic acid; 5-CQA—5-O-caffeoylquinic acid; FER—ferulic acid; 3,5-diCQA—3,5-dicaffeoylquinic acid; 4,5-diCQA—4,5-dicaffeoylquinic acid; 3,4-diCQA—3,4-dicaffeoylquinic acid; 3-CQA—3-O-caffeoylquinic acid; CAF—caffeine; 4-CQA—4-O-caffeoylquinic acid; Nicotinamide, NicAcid—nicotinic acid; QUI—quinic acid; THP—theophylline; THB—theobromine).

**Figure 10 antioxidants-12-00095-f010:**
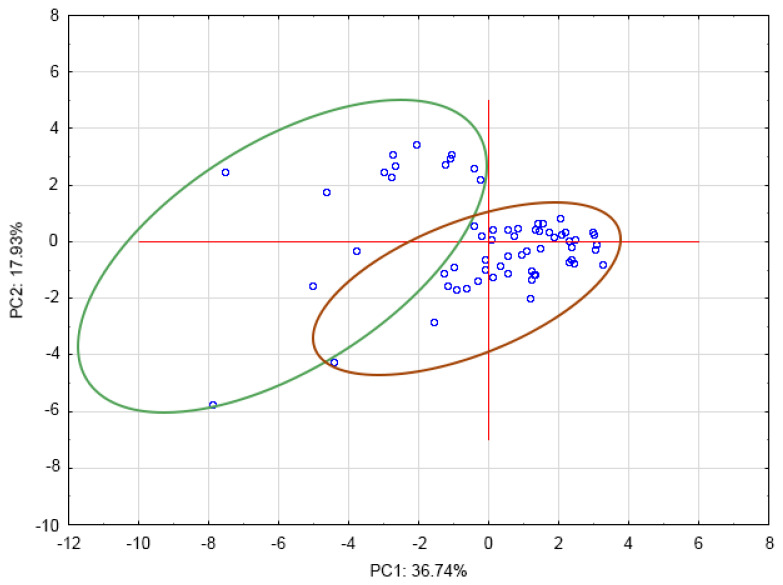
Principal Component Analysis: PC1 × PC2–clusters (the green oval represents green coffee samples and the brown oval represents all processed coffees: roasted–light, medium and dark, steamed, decaffeinated, black beans Robusta and Arabica Kopi Luwak samples).

**Table 1 antioxidants-12-00095-t001:** Abbreviations, origins, and detailed information of *Coffea arabica* and *C. robusta* beans.

Abbr.	Origin of Coffee	Washed/Unwashed	Detailed Information
** *Coffee arabica* **			
ABM	Brazil São Paulo, region: Mogiana	Unwashed coffee	Mogiana TF-–type New York 2/3 Fancy–no black beans allowed
ABMT	Brazil MTGB	Unwashed coffee	MTGB—medium to good beans; Flat beans-screen 15/16
AC	Colombia	Fully washed coffee	Excelso, screen 15
AG	Guatemala	Washed coffee	SHB—strictly hard beans 1600/1700 m The “Strictly Hard Bean” coffees—the best in the world: complete, full-bodied taste, acid and fragrant cup.
AP	Papua New Guinea	Washed coffee	Grows at altitudes of 1000 m,
AUB	Uganda, region: Bugishu	Fully washed coffee	1300–2300 m above sea Grows mostly on the high slopes of Mount Elgon—to the East (the Sipi region), along the Kenyan border
AUD	Uganda, region: Drugar	Unwashed coffee	West—alongside the mountain border with Congo
AL	Bali, Indonesia	Natural fermentation in animal gut	Kopi Luwak Degree of roasting: city–full city
** *Coffee robusta* **			
RBC1	Brazil Conilon 5/6	Unwashed coffee	Type 5/6—screen 13 small beans; max 100 defects
RBC2	Brazil Conilon 6/7	Unwashed coffee	Type 6/7; max 200 defects
RT	Tanzania sc. 14	Unwashed coffee	Screen 14—maximum 5% below screen 14 Dried in cherry in the sun and later milled to remove the outer hard skin and the husks
RU	Uganda sc. 12	Unwashed coffee	900–1500 meters in a radius around Lake Victoria as well as in the western regions. Screen 12—dry-processed beans—no more than 20% of defective beans and not more than one-tenth of 1% by the weight of the extraneous matter shall have a maximum moisture content of 12.5%
RUBB	Uganda BB	Unwashed coffee	900–1500 meters in a radius around Lake Victoria as well as in the western regions; BB—Black Beans—black and discolored coffee beans, separated from clean coffee electronically or by hand
RV	Vietnam sc. 16 clean	Unwashed coffee	Grade 1: > 7 mm bean size
RVD	Vietnam sc. 16 clean decaf	Unwashed coffee	Grade 1: > 7 mm bean size, decaffeinated coffee beans
RVBB	Vietnam gr. 2, 5% BB	Unwashed coffee	Grade 2: Black beans and broken 2–3%, bean size > 6–7 mm
RVBBS	Vietnam gr. 2, 5% BB, Steamed	Unwashed coffee	Grade 2: Black beans and broken 2–3%, bean size > 6–7 mm Steamed beans

**Table 2 antioxidants-12-00095-t002:** Theanine, melatonin, and serotonin content in Arabica and Robusta coffee brews (µg g^−1^).

Coffee ^1^	Unroasted–Green (G) /Roasting Degree (R1–R3; Light, Medium, Dark)	Theanine	Melatonin	Serotonin
* **Coffea arabica** *				
ABM	G	5.27 ± 0.21	0.0006 ± 0.0001	15.48 ± 0.13
	R1	0.09 ± 0.01	0.0030 ± 0.0003	0.13 ± 0.02
	R2	0.16 ± 0.02	0.0051 ± 0.0004	0.05 ± 0.01
	R3	0.12 ± 0.02	0.0027 ± 0.0002	0.09 ± 0.02
ABMT	G	3.64 ± 0.09	0.0009 ± 0.0001	9.96 ± 0.11
	R1	0.06 ± 0.01	0.0036 ± 0.0003	0.13 ± 0.03
	R2	0.08 ± 0.01	0.0038 ± 0.0003	0.03 ± 0.01
	R3	0.15 ± 0.02	0.0072 ± 0.0005	0.09 ± 0.01
AC	G	3.57 ± 0.09	0.0014 ± 0.0001	20.67 ± 1.01
	R1	0.05 ± 0.01	0.0028 ± 0.0003	0.20 ± 0.02
	R2	0.03 ± 0.01	0.0019 ± 0.0002	0.11 ± 0.01
	R3	0.29 ± 0.03	0.0022 ± 0.0002	0.10 ± 0.01
AG	G	3.87 ± 0.09	0.0006 ± 0.0001	22.41 ± 1.21
	R1	0.09 ± 0.01	0.0015 ± 0.0002	0.25 ± 0.03
	R2	0.03 ± 0.01	0.0064 ± 0.0005	0.16 ± 0.02
	R3	0.11 ± 0.01	0.0053 ± 0.0005	0.10 ± 0.01
AP	G	4.68 ± 0.12	0.0009 ± 0.0001	33.26 ± 1.13
	R1	0.04 ± 0.01	0.0019 ± 0.0002	0.23 ± 0.02
	R2	0.06 ± 0.01	0.0019 ± 0.0002	0.13 ± 0.01
	R3	0.10 ± 0.01	0.0023 ± 0.0002	0.10 ± 0.01
AUB	G	2.08 ± 0.08	0.0016 ± 0.0002	13.67 ± 0.74
	R1	0.04 ± 0.01	0.0018 ± 0.0002	0.16 ± 0.01
	R2	0.03 ± 0.01	0.0014 ± 0.0001	0.10 ± 0.01
	R3	0.03 ± 0.01	0.0020 ± 0.0002	0.11 ± 0.01
AUD	G	6.34 ± 0.13	0.0024 ± 0.0002	31.15 ± 1.23
	R1	0.06 ± 0.01	0.0034 ± 0.0002	0.15 ± 0.03
	R2	0.11 ± 0.01	0.0018 ± 0.0001	0.13 ± 0.01
	R3	0.18 ± 0.02	0.0016 ± 0.0001	0.09 ± 0.01
AL	R2	0.06 ± 0.01	0.0038 ± 0.0002	0.06 ± 0.01
* **Coffea robusta** *				
RBC1	G	6.60 ± 0.17	0.0014 ± 0.0002	16.22 ± 0.97
	R1	0.15 ± 0.01	0.0038 ± 0.0003	0.13 ± 0.01
	R2	0.13 ± 0.01	0.0120 ± 0.0009	0.07 ± 0.01
	R3	0.18 ± 0.02	0.0039 ± 0.0002	0.07 ± 0.01
RBC2	G	8.73 ± 0.41	0.0039 ± 0.0002	28.11 ± 1.21
	R1	0.09 ± 0.01	0.0040 ± 0.0002	0.11 ± 0.01
	R2	0.40 ± 0.08	0.0044 ± 0.0003	0.12 ± 0.01
	R3	0.08 ± 0.01	0.0026 ± 0.0002	0.10 ± 0.01
RT	G	2.75± 0.11	0.0072 ± 0.0003	18.52 ± 0.87
	R1	0.20 ± 0.01	0.0025 ± 0.0002	0.21 ± 0.02
	R2	0.12 ± 0.01	0.0039 ± 0.0002	0.13 ± 0.01
	R3	0.11 ± 0.01	0.0052 ± 0.0003	0.11 ± 0.01
RU	G	7.39 ± 0.18	0.0053 ± 0.0002	19.52 ± 0.99
	R1	0.11 ± 0.01	0.0084 ± 0.0004	0.22 ± 0.01
	R2	0.08 ± 0.01	0.0060 ± 0.0003	0.16 ± 0.02
	R3	0.63 ± 0.06	0.0035 ± 0.0002	0.11 ± 0.01
RUBB	G	10.11 ± 0.19	0.0043 ± 0.0002	16.00 ± 0.87
	R1	0.91 ± 0.02	0.0036 ± 0.0002	0.08 ± 0.01
	R2	0.14 ± 0.01	0.0017 ± 0.0002	0.10 ± 0.02
	R3	1.34 ± 0.01	0.0039 ± 0.0002	0.09 ± 0.02
RV	G	9.61 ± 0.21	0.0051 ± 0.0003	19.33 ± 0.98
	R1	0.65 ± 0.03	0.0092 ± 0.0004	0.09 ± 0.01
	R2	0.39 ± 0.03	0.0021 ± 0.0002	0.23 ± 0.01
	R3	0.19 ± 0.02	0.0065 ± 0.0004	0.12 ± 0.01
RVD	G	7.79 ± 0.20	0.0025 ± 0.0002	0.90 ± 0.08
	R1	0.26 ± 0.03	0.0044 ± 0.0003	0.04 ± 0.01
	R2	0.16 ± 0.02	0.0031± 0.0002	0.04 ± 0.01
	R3	0.15 ± 0.02	0.0024 ± 0.0002	0.04 ± 0.01
RVBB	G	8.63 ± 0.24	0.0061 ± 0.0005	19.67 ± 1.12
	R1	0.12 ± 0.02	0.0028 ± 0.0002	0.25 ± 0.03
	R2	0.08 ± 0.01	0.0054 ± 0.0004	0.22 ± 0.03
	R3	0.40 ± 0.01	0.0038 ± 0.0003	0.17 ± 0.02
RVBBS	G	0.27 ± 0.03	0.0018 ± 0.0002	0.20 ± 0.02
	R1	0.10 ± 0.01	0.0015 ± 0.0002	0.05 ± 0.01
	R2	0.14 ± 0.01	0.0035 ± 0.0003	0.08 ± 0.01
	R3	0.19 ± 0.02	0.0024 ± 0.0002	0.05 ± 0.01

^1^ Abbreviations and information about the coffee was provided in Table 1.

## Data Availability

Data is contained within the article and Appendix A.

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
