# Peer review of "Comprehensive Analysis of Metabolites in Brews Prepared from Naturally and Technologically Treated Coffee Beans"

_antioxidants, 2022, doi:10.3390/antiox12010095_

Round 1
Reviewer 1 Report
In this manuscript, the Auhors describe the results obtained using LC-MS/MS to analyse the content of chlorogenic acid isomers, alkaloids, nicotinic acid and nicotinamide, serotonin and melatonin as well as organic acids of brews from different coffee beans, in order to assess the effects of variety, cultivation location and processing on brew composition. The work was conducted with care and the manuscript is well written. I suggest publication after some minor modifications. Specific comments are as follows:
Title: I suggest to omit “including including Kopi Luwak coffee brew”, since this information is present in the text.
Abstract, line 22: please omit “regular”. Please explain the most important differences induced by technological treatment (lines 25-26: what does this “influence” involve?),
Materials and methods, line 125: please specify the dilution (solvent and ratio)
Results: please add error bars in Figures
3.6: please change “projection” to “the first two components biplot”. It explains a 54.67% of total variance. How many PCs were extracted? Was the PCA conducted on the correlation matrix?
Line 395: it does not seem that PC1 negativaly correlates with all variables.
Line 396: please explain “important”.
Author Response
Response to Reviewers Comments
Thank you for your helpful comments and suggestions.
The text has been revised according to the recommendations which certainly increased the value of this work. Please find the responses attached to the comments point by point.
All changes made in the revised manuscript are highlighted in red color.
Reviewer 1
In this manuscript, the Auhors describe the results obtained using LC-MS/MS to analyse the content of chlorogenic acid isomers, alkaloids, nicotinic acid and nicotinamide, serotonin and melatonin as well as organic acids of brews from different coffee beans, in order to assess the effects of variety, cultivation location and processing on brew composition. The work was conducted with care and the manuscript is well written. I suggest publication after some minor modifications. Specific comments are as follows:
Title: I suggest to omit “including Kopi Luwak coffee brew”, since this information is present in the text.
We have deleted the information from the title as suggested.
Abstract, line 22: please omit “regular”. Please explain the most important differences induced by technological treatment (lines 25-26: what does this “influence” involve?),
We have changed regular to washed and unwashed coffee beans. The influence is connected with decreasing or increasing amount of the compound
Materials and methods, line 125: please specify the dilution (solvent and ratio)
Dilution depends on the quantity of determined compounds in green or roasted Arabica or Robusta coffee brews. We have added an appropriate description.
Results: please add error bars in Figures
We have added the SD as suggested.
3.6: please change “projection” to “the first two components biplot”. It explains a 54.67% of total variance. How many PCs were extracted? Was the PCA conducted on the correlation matrix?
We have changed as suggested. 16 PCs were extracted. PCA was conducted on the correlation matrix.
Line 395: it does not seem that PC1 negativaly correlates with all variables.
We have changed as suggested.
Line 396: please explain “important”.
The most important compounds possess the largest coefficients (in absolute value) close to 1, such as 5-CQA = -0.936; 3,4-diCQA = -0.687; p-coumaric acid = -0.737; caffeic acid = 0.729; 3-CQA = -0.562; ferulic acid = -0,536 with high antioxidant activity

Reviewer 2 Report
The paper described the composition in metabolites of different brews prepared under different technological treatment in addition to untreated and Kopi Luwak coffees.
It is generally well writtend and clear.
However, considering the topic of the special issue, references to the antioxidant activity should be added in the introduction section and the obtained results should be correlated to the same activity on literature base and better discussed.
In addition, some minor concernings: for example Line 62: eliminate "(Farah and de Paula Lima, 2019a)." after [11]
Author Response
Response to Reviewers Comments
Thank you for your helpful comments and suggestions.
The text has been revised according to the recommendations which certainly increased the value of this work. Please find the responses attached to the comments point by point.
All changes made in the revised manuscript are highlighted in red color.
Reviewer 2
The paper described the composition in metabolites of different brews prepared under different technological treatment in addition to untreated and Kopi Luwak coffees.
It is generally well written and clear.
However, considering the topic of the special issue, references to the antioxidant activity should be added in the introduction section and the obtained results should be correlated to the same activity on literature base and better discussed.
We have added information about antioxidant activity in Introduction and Discussion section.
In addition, some minor concernings: for example Line 62: eliminate "(Farah and de Paula Lima, 2019a)." after [11]
We have deleted the names after the Reference number.

Round 2
Reviewer 2 Report
Considering the improvement of the manuscript, it is acceptable for the publication in the present form.